# Automated Patch-Clamp and Induced Pluripotent Stem Cell-Derived Cardiomyocytes: A Synergistic Approach in the Study of Brugada Syndrome

**DOI:** 10.3390/ijms24076687

**Published:** 2023-04-03

**Authors:** Dario Melgari, Serena Calamaio, Anthony Frosio, Rachele Prevostini, Luigi Anastasia, Carlo Pappone, Ilaria Rivolta

**Affiliations:** 1Institute of Molecular and Translational Cardiology (IMTC), San Donato Milanese, 20097 Milan, Italy; 2Faculty of Medicine, Vita-Salute San Raffaele University, 20132 Milan, Italy; 3Arrhythmology Department, IRCCS Policlinico San Donato, San Donato Milanese, 20097 Milan, Italy; 4School of Medicine and Surgery, University of Milano-Bicocca, Via Cadore, 48, 20900 Monza, Italy

**Keywords:** Brugada syndrome (BrS), BrS experimental model, automated patch-clamp, human-induced pluripotent stem cells, hiPSCs, hiPSCs-CM, patient-derived cardiomyocytes, personalized medicine

## Abstract

The development of high-throughput automated patch-clamp technology is a recent breakthrough in the field of Brugada syndrome research. Brugada syndrome is a heart disorder marked by abnormal electrocardiographic readings and an elevated risk of sudden cardiac death due to arrhythmias. Various experimental models, developed either in animals, cell lines, human tissue or computational simulation, play a crucial role in advancing our understanding of this condition, and developing effective treatments. In the perspective of the pathophysiological role of ion channels and their pharmacology, automated patch-clamp involves a robotic system that enables the simultaneous recording of electrical activity from multiple single cells at once, greatly improving the speed and efficiency of data collection. By combining this approach with the use of patient-derived cardiomyocytes, researchers are gaining a more comprehensive view of the underlying mechanisms of heart disease. This has led to the development of more effective treatments for those affected by cardiovascular conditions.

## 1. Introduction

Brugada syndrome (BrS) is a rare disorder with a worldwide prevalence of 5–20 cases per 10,000 people in the population, but these numbers are likely an underestimation of the true extent of the phenomenon. Individuals with BrS are at an elevated risk of sudden cardiac death (SCD), particularly between the ages of 30 and 50 years, due to ventricular fibrillation or polymorphic ventricular tachycardia even in absence of structural heart defects [1]. Despite ongoing efforts, the genetics of BrS remain mostly unknown, hindering diagnosis and risk stratification. While implantable cardiac defibrillator (ICD) therapy is currently the most effective strategy to prevent recurrent ventricular fibrillation, which is the leading cause of sudden death, it carries a high risk of lifetime complications, making risk–benefit analysis difficult. In contrast, catheter ablation over the anterior aspect of the right ventricular outflow tract epicardium has shown efficacy in controlling ventricular arrhythmias, reducing the incidence of BrS electrocardiographic patterns, and lowering the number of ICD shocks. Researchers have focused on both clinical studies to improve patient stratification, and translational research to deepen the understanding of the pathogenesis and development of arrhythmic events in BrS. For years, BrS has been considered a monogenic disease, mostly associated with mutations in the sodium voltage-gated channel alpha subunit 5 (*SCN5A*) gene encoding the voltage-gated cardiac sodium channel Na_V_1.5 [1,2]. Currently, increasing evidence, together with the discovery of new variants of uncertain significance (VUS) [3], suggest that BrS is a combination of multiple genetic factors and environmental features that act in concert for the onset and the severity of the disease. In fact, *SCN5A* mutations only account for 20% of patients diagnosed with the syndrome [4,5]. In the past decade, the study of the most common clinical group of patients, i.e., those diagnosed with BrS who do not exhibit typical mutations in Na_V_1.5, has gained significant attention [3,6]. Therefore, in the field of translational research, classical monogenic approaches that involve the characterisation of single variants in heterologous expression systems may not be the ideal choice.

Notably, a recent study [7] challenged the notion of BrS as a genetic disease that solely impacts the human heart, linking the disease to noncardiac conditions such as epilepsy, thyroid disorders, cancer, skeletal muscle sodium channelopathies, laminopathies, and diabetes. This suggests that, apart from its involvement in the heart, BrS may also affect other parts of the body, including the nervous system, digestive system, and metabolic processes. For these reasons, researchers interested in this field are facing a historical moment, which is seeing an explosion of new potential targets, that are at least involved in, if not responsible for, the pathogenesis of BrS. In this sense, the combination of the most updated single-cell electrophysiological applications with a high-performance technique such as automated patch-clamp (APC), with biotechnological advancements such as those brought by human inducible pluripotent stem cells (hiPSCs), creates the most suitable environment and represents a breakthrough in the study of such a complex disease.

## 2. Automated Patch-Clamp Electrophysiology

### 2.1. The Rise of Automation in Cellular Electrophysiology

APC was developed at the turn of the new millennium, introducing a new approach to the study of ion channels. APC boasts its strength in producing high-throughput data, at an unprecedented level in the field of electrophysiology. This allows for even untrained personnel to acquire vast amounts of high-quality data in a relatively short time, whereas manual patch-clamp (MPC) requires the expertise of highly specialised operators who patch cells one by one in lengthy and often low-yield experimental sessions. Although APC data may not be as refined or versatile as MPC data, when these devices first emerged, different companies explored various methods to achieve automated, high-quality recordings. One such example is an automated eight-channel two-electrode voltage-clamp system for *Xenopus* oocytes that was developed and commercialised by Axon Instruments [8]. OpusXpress was a short-lived device and is now discontinued, although some units remain in operation and studies using this machine are still being published [9]. The most successful approach was the chip-based planar array in which single cells in suspension are captured by micron-sized holes placed on the bottom of borosilicate glass- or plastic-based wells [10]. In this method, single wells are filled with an extracellular solution, while an intracellular solution is delivered to small chambers below the well bottom wall. The two chambers are connected through the micron-sized hole, which serves the same function as the pipette opening in conventional MPC. Through automated suction, cells attach to the holes and form a giga-seal, and a further application of negative pressure causes the seal to rupture, enabling stable recordings in the whole-cell configuration [11]. Over the past two decades, APC devices have become more sophisticated, and have increased the number of parallel recordings up to 384, as seen with SyncroPatch 384 from Nanion Technologies GmBH and Qube 384 from Sophion Bioscience [12]. These devices, with their high reliability, automation, and ease of use, are revolutionizing the field of ion channel study, and are used in various applications, including drug discovery, safety testing, and ion channel characterisation (for review, see [10,11,12]), becoming a new gold standard for the pharmaceutical industry.

### 2.2. APC in Pharma and in Academia

The International Council for Harmonisation of Technical Requirements for Pharmaceuticals for Human Use (ICH) is an organisation which brings together regulatory authorities and pharmaceutical industry to discuss scientific and technical aspects of drugs with the goal to develop and release ICH guidelines. Since the release of the ICH S7B standard for clinical testing in 2005, and more recently the ICH E14 standard for clinical trials of new compounds in 2020, the gold standard for ensuring the cardiac safety of new compounds has been the pharmacological profiling of the human ether-a-go-go-related (hERG) channel (K_v_11.1), which is a key regulator of ventricular cardiomyocyte repolarisation. According to these guidelines, the inhibition of the hERG channel is used as the main predictor of QT interval prolongation and proarrhythmogenic risk. Despite their success, these standards have proven to be limited and incomplete by various studies such as those on the d (MICE) model [13], leading to the development of a new approach known as the comprehensive in vitro proarrhythmia assay (CiPA). In conjunction with the United States Food and Drug Administration, this approach involves three key steps: first, new compounds should be tested using patch-clamp techniques on multiple human cardiac ion channels expressed in heterologous cell lines, not just hERG; second, patch-clamp data should be used to predict proarrhythmogenic risk in computational models of human ventricular cardiomyocytes; and finally, these simulations must be experimentally confirmed by tests on cardiomyocytes derived from hiPSC (hiPSC-CMs) [14]. Concerning the first point, in the pharmaceutical industry, APC technology is used to screen large numbers of compounds for potential therapeutic effects on several specific ion channels. This allows the rapid identification of molecules that could be developed into drugs for a variety of diseases. Considering the third point, while iPSC-CMs have many advantages, such as the ability to model human-specific diseases, one limitation is their immaturity in terms of electrophysiology [15], characterised by low levels of the inward rectifying potassium (K^+^) channel (IK_1_). This lack of IK_1_ is a crucial factor in proarrhythmic traits observed in iPSC-CMs. However, recent advances have helped to overcome this limitation (see below). The combination of APC with dynamic clamp technique has allowed for the real-time calculation and electronic injection of the IK_1_ current, resulting in action potentials that resemble those of adult cells [16,17]. Although MPC will never be fully replaced, the integration of dynamic APC with iPSC-CM technology has the potential to contribute to more than one step of the CiPA, which is essential for meeting the high-throughput demands of the pharmaceutical industry and regulatory authorities [12,15].

In the past decade, APC has also found its way into the academic field of channelopathies and it has been used to study the physiology and pharmacology of ion channels. This advancement has been used to characterise mutant ion channels associated with various cardiac and neurological disorders (Table 1). The high throughput of APC allows a relative quick evaluation of large variant libraries, improving functional and clinical interpretation, especially of those of unknown significance [18,19,20]. Additionally, deep mutational scanning (DMS) is a recently developed complementary approach that relies on APC to classify hundreds of disease-related variants [21] which could be beneficial in functional studies of ion channel kinetics and mechanics. However, despite the widespread enthusiasm for APC technology in the scientific community, its high costs of initial purchase and maintenance make it inaccessible to many universities and research labs that operate on tight budgets. As a result, access to this technology is often limited to well-funded research institutions and pharmaceutical companies, which can limit opportunities for academic researchers to use it and contribute to the field.

## 3. Brugada Syndrome Experimental Models

Over the years, various experimental models have been developed for studying BrS-related protein variants, creating BrS-like symptomology, and testing potential treatments, with a particular emphasis on ion channels (Figure 1). Each model has its own unique advantages and disadvantages, and selecting the most suitable one depends on the research question and available resources. The following paragraphs provide a summary of the most commonly used cellular and animal models in BrS research, starting with the simpler and historical and progressing to the more recent and complex ones. It is important to note that the results obtained from one model may not be directly applicable to another.

### 3.1. Heterologous Expression Systems

Heterologous expression systems are used to express foreign genes in host organisms different from their natural host, and they are essential tools for studying proteins in normal and pathological variants related to several diseases. They provide a robust means of characterizing the biophysical properties of ion channels at a single-cell level, and they serve as the benchmark for examining the effects and safety of new compounds intended for the disease treatment [30]. *Xenopus oocytes* were the first cell models applied for electrophysiological analysis of ion channels, receptors, and transporters using the two-electrode voltage-clamp technique [31,32]. Although this method was later replaced by giga-seal patch-clamp techniques [33] applied to immortalised cell lines, *Xenopus* oocytes continue to be used due to their high reliability, ease of maintenance, and ability to be microinjected with messenger ribonucleic acid (mRNA) or deoxyribonucleic acid (DNA) constructs [8]. However, the large size of the oocytes can pose limitations to recording fast-kinetics currents [34], such as the sodium current (I_Na_) involved in BrS. Despite this, functional characterisations of BrS-related ion channel variants in *Xenopus* oocytes are still being published today [35,36].

The most commonly used single-cell models for characterizing BrS-related ion channel variants are human embryonic kidney (HEK293) cells, SV40 transformed (tsA201) cells, Chinese hamster ovary (CHO) cells, and simian fibroblast-like COS-7 cells derived from kidney tissue (see [37] for a comprehensive review). Immortalised cells have proven to be a convenient and cost-effective means to explore BrS due to their ability to provide detailed and high-quality patch-clamp characterisations of ion channel biophysical properties. Nonetheless, these models are not without limitations, as they can experience challenges in achieving optimal expression levels of foreign genes, protein instability, and differences in posttranslational modifications that may result in inconsistent findings between different cell lines, as the physiological environment of a studied channel may not always be accurately replicated [37,38]. Developing and optimizing a heterologous expression system can be a time-consuming and expensive process, and there is always the risk of the system not working as intended. Additionally, single-cell-level models lack the complex BrS-related behaviours seen in the electrocardiogram (ECG).

### 3.2. Zebrafish Model

Zebrafish (*Danio rerio*) has emerged as an excellent animal model for cardiovascular and basic research, owing to its fully characterised genome, fast development, and larval optical transparency, as well as its low maintenance costs, high reproduction rates, and the fact that it has over 70% genetic homology with humans, including up to 84% of human disease-related genes [39]. Its short life cycle of three months, from fertilisation to adulthood, further enhances its appeal as a model organism. Due to the transparency of zebrafish embryos and larvae, normal and induced cardiovascular morphological and physiological changes can be directly observed through conventional microscopy techniques. The electrophysiology of the zebrafish heart is similar to that of humans, with an ECG that closely resembles the human one, and a ventricular action potential (AP) exhibiting the plateau phase seen in humans but absent in mice, as well as a robust IK_1_ current for sustaining resting membrane potential. The functionality of the zebrafish heart has been assessed using conventional patch-clamp techniques, as well as ex vivo and in vivo recordings of whole zebrafish hearts or isolated atrial and ventricular cardiomyocytes [40]. A BrS model in zebrafish was generated by knocking out the glutathione S-transferase mu 3 (*GSTM3*) gene, a genetic modifier of the BrS phenotype [41]. The study confirmed the model’s reliability by demonstrating a significant increase in the PR interval, QRS duration, and the number of ventricular arrhythmic events after flecainide treatment in transgenic zebrafish compared to wild-type (WT) fish, while quinidine infusion returned the values to normal levels. Nonetheless, studies on sudden cardiac death involving zebrafish are still limited.

### 3.3. Mouse Models

Two main methods are commonly used to study Brugada syndrome (BrS) in mice models: generating *SCN5A* knockout (KO) animals and engineering mice with a specific BrS-related genetic mutation found in patients [38]. The first approach was used to create a cardiac arrhythmogenic model [42] in which homozygous KO mice showed embryonic lethality due to heart structural abnormalities, while the ECG of heterozygous KO (*SCN5A*^+/−^) showed a slower conduction velocity and roughly a 50% reduction in sodium (Na^+^) conductance compared to WT mice, which explains various phenotypic features seen in in vivo experiments, such as impaired AP propagation, conduction block, re-entrant arrhythmias, and ventricular tachycardia. However, the cardiac phenotypes observed in *SCN5A*^+/−^ mice were diverse and heterogeneous, and not entirely representative of BrS. The second approach, engineering mice with a specific BrS mutation, was used to create a mouse carrying the murine equivalent mutation (*SCN5A*-1798insD) of the human *SCN5A*-1795insD variant, found in a family who exhibited an overlap syndrome of long QT syndrome 3 (LQT3), BrS and progressive cardiac conduction defects [43]. *SCN5A*-1798insD mice displayed a similar phenotype, proving that a single mutation was sufficient to cause the overlap syndrome. Interestingly, the severity of the conduction defect caused by the *SCN5A*-1798insD/+ was shown to be strain-dependent [44].

The use of murine models, compared to heterologous systems, allows the investigation of mutation-related effects in the whole organism and through its development, parallel to the possibility to study native currents and AP in isolated cardiomyocytes. However, differences in ion channel expression patterns and AP features between mice and humans, which can result also in different pharmacology, must be considered.

### 3.4. Rabbit Models

Just like mice, rabbits are small and inexpensive animals with a short reproductive cycle, and their cardiac electrophysiology and ion channel expression profile are more similar to humans than mice. However, the pharmacological induction of the BrS-like phenotype is necessary. Researchers induced J-wave elevation and spontaneous ventricular fibrillation in Langendorff-perfused rabbit hearts through the activation of small-conductance calcium-activated K^+^ channels and inhibition of Na^+^ channels using cyclohexyl-[2-(3,5-dimethyl-pyrazol-1-yl)-6-methyl-pyrimidin-4-yl]-amine (CyPPA) or hypothermia. Moreover, CyPPA impaired atrioventricular node and intraventricular conduction. This induced condition resembles BrS, which often includes bradycardia and conduction delay [38,45]. A separate study used the transient outward potassium (I_to_) activator NS5806 to induce a BrS-like ECG in Langendorff-perfused rabbit hearts, but no sustained arrhythmias or phase-2 re-entry were observed, indicating that this framework does not fully replicate BrS arrhythmogenesis in intact rabbit hearts [46].

### 3.5. Porcine Models

The pig heart is similar to the human heart in terms of rate, size, anatomy, and autonomic innervation [47]. Researchers engineered a porcine model carrying the equivalent of a human nonsense mutation (*SCN5A*-E558X) found in a child with BrS [48]. The heterozygous animals showed reduced expression of the Na_v_1.5 protein and exhibited conduction abnormalities, in absence of cardiac structural defects. Sudden cardiac death was not observed in piglets up to 2 years of age, but the *SCN5A*^E558X/+^ hearts were highly arrhythmic when Langendorff-perfused. Although the model replicated many features of the human condition, it did not develop a BrS-patterned ECG, even with high-dose flecainide. This may be due to the absence of the calcium-independent I_to_ current in the pig heart. Additionally, the high costs and long reproductive cycle of pigs limit their usefulness in studying BrS.

### 3.6. Canine Models

Dogs are commonly used as a model for studying cardiac electrophysiology and, like rabbits, the BrS pattern in their ECG has to be pharmacologically induced [49,50,51]. These drugs, such as the Na^+^ channel blocker pilsicainide, the antihistamine and hERG channel blocker terfenadine, and the ATP-sensitive K^+^ channel agonist pinacidil, can be perfused into isolated right ventricular tissue preparations [52]. The I_to_ activator NS5806, which failed to induce BrS in a rabbit model [46], has been shown to cause electrographic and arrhythmic manifestations typical of BrS [53]. These drug-inducible systems have been instrumental in uncovering several arrhythmia-related ECG landmarks including ST-segment elevation [54], J-wave/Osborne wave [55] and T-wave alternans and ventricular tachycardia or fibrillation [56]. However, caution must be taken when extrapolating pharmacological studies to a clinical setting as they may not reproduce the clinically observed genotypes. [57].

### 3.7. Native Human Cardiomyocytes

Human cardiomyocytes from patients are the most comprehensive cellular models for studying BrS, but they are also the most challenging and expensive. In a recent study, Schmidt and colleagues investigated the electrophysiology of primary cardiomyocytes obtained from the right atrium of a 64-year-old male patient carrying the N167K mutation in the Na_V_1.5 binding region of β-2-syntrophin [58]. Voltage-clamp recordings showed a reduced Na_V_1.5 peak, late current density and shortened AP compared to control samples. However, the limitations of this study being performed with one single patient cells include the use of atrial cardiomyocytes instead of ventricular ones, as well as the lack of appropriately matched control.

### 3.8. Human-Induced Pluripotent Stem Cell-Derived Cardiomyocytes

In 2006, Shinya Yamanaka and colleagues firstly developed the technique to reprogram somatic cells in induced pluripotent stem cells (iPSCs), using four transcription factors (Octamer-binding transcription factor 3/4, Sex-determining region Y-box 2, Krüppel-like factor 4, and the myelocytomatosis oncogene product also known as Oct3/4, Sox2, Klf4 and c-Myc, respectively), opening up new horizons for research and medicine [59]. The availability of autologous cells that can differentiate into all cell types directly from patients has allowed the possibility of investigating the aetiology of BrS directly on patients’ cardiomyocytes without the need for invasive biopsies to obtain primary cells. These cells provide a sufficient amount of biological material that can be sustained in vitro for a long time, with a more complex and physiological expression profile compared to that of immortalised cell lines and free of species-to-species differences inherent in animal-derived models [60]. Davis and colleagues were among the pioneers in the adoption of iPSC technology for the study of BrS. They derived iPSCs from a patient with a heterozygous *SCN5A* mutation, which presented a mixed phenotype of both BrS and LQTS3. The team then differentiated and functionally characterised iPSC-CMs from the same patient [61]. They also compared different BrS experimental models, and showed that mouse and hiPSC-CMs with equivalent *SCN5A* heterozygous mutations exhibited comparable reductions in peak Na^+^ current density and similar alterations in AP parameters. Additionally, they observed differences between hiPSCs and the HEK293 heterologous expression system, which are likely due to the lack of heterozygous channel expression in the HEK293 model, and the presence of only one beta subunit instead of the four isoforms expressed in hiPSC-CMs. This study demonstrated that iPSC-CMs can be considered a suitable model to reproduce Na_V_1.5 channel pathophysiology.

Since then, many other iPSC lines have been derived from patients with mutations in the *SCN5A* gene (see Table 2 for a complete list). However, it is important to note that the gene expression and functions of hiPSC-CMs are more similar to those of embryonic or foetal cardiomyocytes than to adult ones. In fact, hiPSC-CMs lack T-tubules, have underdeveloped contractile and calcium release machinery, are missing the IK_1_ current, and display prominent pacemaker currents [62,63]. This may pose some limitations to studying a complex disease like BrS. In some cases, the immaturity of hiPSC-CMs can conceal the pathological phenotype, as observed by Okata and colleagues in 2016 [64]. Their study showed that the embryonic type Na^+^ channel beta subunit (*SCN3B*) masked the BrS traits in iPSC-CMs derived from a patient with the E1784K-*SCN5A* mutation. The reduced peak I_Na_ density was successfully revealed by a knockdown of *SCN3B* in BrS hiPSC-CMs.

In this sense, several methods have been employed to improve the maturation stage of iPSC-CMs and increase the expression of the adult form of the *SCN5A* gene. These include long-term culture, culture on a rigid substrate [65,66], electrical and mechanical stimulation [67,68], and the use of biochemical cues such as the growth hormone Tri-iodo-L-thyronine [69]. These advancements refined the iPS-CM technique and have allowed the production and characterisation of iPSCs leading to a deeper understanding of different ion channels involved in BrS, including L-type Ca^2+^ channels [70,71], the tetrodotoxin (TTX)-resistant Na^+^ voltage-gated channel alpha subunit 10 (*SCN10A*) [72,73], and K^+^ channels including the K^+^ voltage-gated channel subfamily Q member 1 (*KCNQ1*) and the K^+^ voltage-gated channel subfamily H member 2 (*KCNH2*) [3,74].

A recent study found that a tricellular three-dimensional cardiac microtissue approach improved the switch to the adult *SCN5A* isoform. This method was used to characterise a *SCN5A* exon 6B mutation in iPSC-CMs derived from a patient with tachycardia and conduction disorder [75]. Indeed, patient-derived iPSC-CMs provide a more comprehensive physiological model, elevating the level of functional characterisation complexity with respect to the traditional variant description through protein overexpression. For instance, Liao and colleagues studied the impact of inflammation on arrhythmogenesis in BrS by incubating patient-derived hiPSC-CMs harbouring a loss-of-function (LoF) mutation in the *SCN10A* gene with lipopolysaccharides. The data showed that after 48 h of incubation, the patient-derived cells displayed a decrease in the already impaired Na^+^ peak current density and small but significant changes in channel kinetics compared to control cells derived from three healthy donors. Additionally, the effect was prevented by the protein kinase C (PKC) inhibitor chelerythrine, indicating that inflammation may worsen LoF Na^+^ variants in BrS patients through enhanced reactive oxygen species (ROS)-PKC signalling [73]. In a separate study, Belbachir and colleagues identified a variant in the Ras associated with diabetes (RAD) guanosine triphosphatases (GTPase) gene (*RRAD*) in a BrS patient, derived iPSC-CMs and found a reduction in the velocity of the AP upstroke, prolonged AP duration, and increased probability of early afterdepolarisations (EADs), as well as a reduction in Na^+^ and Ca^2+^ peak current densities and increased late I_Na_. Structural analysis unveiled a reduction in focal adhesion and abnormal actin filament distribution, resulting in three-dimensional structural aberrations [76].

A significant advantage of using hiPSC-CMs in the study of genetic disorders is that they retain the complete genotype of the patient, including all the genetic variants that may contribute to the disease. For example, El-Battrawy and colleagues found a patient with BrS harbouring two variants in the ancillary β1 subunit (*SCN1B*) of the Na^+^ channel. The iPSC-CMs derived from the patient’s skin fibroblasts recapitulated the BrS phenotype, showing a reduction in the peak density of the I_Na_, and alterations in channel activation, inactivation, and recovery from inactivation kinetics, leading to a reduced depolarisation velocity and amplitude in AP. Additionally, the patient-derived iPSC-CMs exhibited increased arrhythmic events compared to healthy controls [77]. Another study by Barajas-Martinez and coworkers [78] found that iPSC-CMs derived from a patient carrying three individual mutations in the ryanodine receptor-associated FK506-binding protein 1B (FKBP1b), in the voltage-gated Na^+^ channel *SCN9A* (which encodes the voltage gated neuronal Na^+^ channel Na_V_1.7), and peroxidasin-like protein PXDNL, showed a reduction in the Ca^2+^ but not in the Na^+^ current, as well as irregular spontaneous activity, which replicated the patient’s clinical phenotype. However, the patient’s asymptomatic relatives who carried only one or two of the mutations did not show any pathological alterations in their iPSC-CMs.

When it comes to modelling diseases through the use of hiPSCs, the availability of isogenic pairs possessing the same genetic background is a great advantage. Isogenic lines can be generated by introducing certain mutations linked to the diseases to normal hiPSCs, or by correcting a mutation in the patient-derived iPSCs, without making any other genetic alterations. This process considerably reduces genetic variability, thus allowing a clearer genotype–phenotype correlation. In fact, family members, sometimes used as a control, are not genetically identical to the proband and discrepancies in single nucleotide polymorphisms (SNPs)—may have an influence on, or even determine, diseased phenotypes. In this perspective, the integration of BrS-derived iPSC-CMs with clustered regularly interspaced short palindromic repeat (CRISPR/Cas9) technology holds great potential, as described by several reviews (see [79,80]). A recent study by Bersell and colleagues highlights the utility of this combination. The authors identified a BrS-related variant in the cardiac T-box transcription factor 5 (*TBX5*) gene, causing a reduction in the I_Na_ peak density and an enhancement of the Na^+^ late current, leading to changes in AP parameters and the increased probability of arrhythmic events. Using CRISPR/Cas9, the authors were able to correct the mutations in patient-derived iPSC-CMs, reverting the diseased phenotype. This work highlights the role of the *TBX5* mutation in BrS and the reversibility of the phenotype in hiPSC-CMs. The study also showed that the same pathological phenotype could be replicated in hiPSC-CMs from healthy controls with the introduction of the same mutation [81]. In another paper, De La Roche and coauthors modified healthy hiPSC lines by introducing the point mutation A735V to the *SCN5A* gene, which had been reported by four separate clinical centres around Europe, America, and Japan. The generated hiPSC-CMs showed an increase in both spontaneous and induced ventricular-like AP duration and upstroke maximal velocity, due to a decrease in I_Na_ density without a reduction in mRNA and protein expression. Additionally, the mutation was found to cause a rightward shift of the channel activation curve and a slower recovery from inactivation [66]. These results reveal the causative role of the Na_v_1.5 variant regardless of the genetic background of a patient, removing the possible heterogeneity coming from individual-specific polymorphism patterns.

In the context of BrS being a multifactorial syndrome caused by a combination of genetic and environmental factors [82], the ultimate goal of patient-derived iPSC-CMs is to study and develop personalised medicine strategies. A recent preclinical study employed the “disease in the dish” approach to investigate the case of an 18-year-old male patient with a gain-of-function (GoF) mutation in the K^+^ voltage-gated channel subfamily D member 3 (*KCND3*) gene, responsible for the cardiac I_to_ [74]. The patient was diagnosed with J-wave syndrome-early repolarisation subtypes, and suffered from a range of cardiac complications including polymorphic ventricular tachycardia, ventricular and atrial fibrillation, atrial flutter, and cardiac arrest. Despite receiving conventional treatments, including quinidine therapy, premature ventricular contraction-targeted ablation, and mexiletine therapy, the patient remained unresponsive. Flecainide treatment was administered and hiPSC-CMs were generated for potential therapeutic testing in a compassionate use scenario. The natural flavone, acacetin, was tested on the hiPSC-CMs at a concentration of 10 µM and was found to restore normal I_to_ levels by reducing the peak current by approximately 40%. This resulted in the effective reversal of the GoF abnormal AP [74]. These findings suggest that acacetin may represent a promising therapeutic agent for this specific patient with a GoF mutation in the *KCND3* gene. Finally, in a recent study by Zhong and colleagues [70], a cellular model of BrS was established using a combination of patient-derived iPSC-CMs, healthy donor-derived iPSC-CMs, and CRISPR/Cas9-engineered isogenic control site variant corrected cells. The model was based on a VUS in the Ca^2+^ voltage-gated channel auxiliary subunit β2 (*CACNB2*) gene, and the study aimed to evaluate the pharmacological effects of different compounds on the cellular model. The *CACNB2* VUS was identified in a 55-year-old patient with recurrent sudden cardiac arrest who received an ICD that discharged appropriately after beta-blocker therapy was discontinued. The patient was then treated with a low dose of bisoprolol, despite the previous literature suggesting that it should be avoided in BrS, and did not experience any ICD discharges over a 5-year follow-up period. Cellular electrophysiological experiments revealed that the peak current density of the L-type Ca^2+^ current (I_CaL_) was smaller in patient-derived iPSC-CMs compared to healthy and isogenic controls, resulting in abnormalities in AP parameters and an increased incidence of arrhythmic events such as EADs. Low-concentration bisoprolol abolished arrhythmic events and reduced beat-to-beat interval variability, supporting the effectiveness of the low-dose therapy used in the patient. These findings suggest that patient-derived iPSCs can be used in preclinical studies to evaluate the effects of drugs that are traditionally avoided in BrS, which can lead to personalised treatment strategies based on gene variants. By highlighting the importance of gene variants in pharmacological evaluation, the study underscores the utility of hiPSC-CMs in advancing our understanding of disease mechanisms and developing personalised therapies.

The implementation of personalised electrophysiological characterisation, which could lead to patient- or gene-specific pharmacological treatments, demands a high level of data generation and analysis that traditional electrophysiological techniques cannot provide. In the following paragraphs, we will examine the potential breakthrough represented by automated patch-clamp technology and its application in studying BrS in combination with hiPSC-CMs.

**Table 2 ijms-24-06687-t002:** BrS-related hiPS-CMs models. (ND = not defined; AC = arrhythmogenic cardiomyopathy; ERS = early repolarisation syndrome; JWS = J-wave syndrome; het = heterozygous; homo = homozygous; HOS = holt-oram syndrome; / = not available).

hiPSC Name	Mutation					Pathology	Derivation	Reprogramming Vector	Method of Modification	Isogenic	Electro Physiology	Drugs	Ref.
	Gene	Coding DNA	RNA	Protein	Zygosity								
SCN5A-het	*SCN5A*	c.5537insTGA	/	p.1795insD	Het	BrS/ LQTS3	47-yo male	Lentivirus	/	/	MPC	/	[61]
									/	/	MPC	GS967 (300 nmol/L)	[83]
iSCN5A									/	/	MPC	/	[84]
PKP2-Ipsc	*PKP2*	c.2484C>T; c.1904G>A	r.[2483_ 2489del]	p.G828G; p.R635Q	Homo	BrS/AC	44-yo female	Retrovirus	Lentivirus Infection in iPSC-CM	AC-Hipsc (PKP2-iPSC JK#11)	MPC	/	[85,86]
hiPSC^R1638X^	*SCN5A*	c.4912C>T	/	p.R1638X	Het	BrS	34-yo male	Sendai Virus	/	/	MPC	PTC124 (17 mmol/L); Gentamicin (20 mmol/L)	[87]
BrS1	*SCN5A*	c.2053G>A; c.2626G>A	/	p.R620H; p.R811H	Het	BrS	44-yo male	Sendai Virus	/	/	MPC	/	[88]
BrS2	*SCN5A*	c.4190delA	/	p.K1397Gfs	Het	BrS	53-yo male	Sendai Virus	CRISPR-Cas9; piggyBac	BrS2-GE iPSC-CMs	MPC	/	
BrS1	ND	/	/	/	/	BrS	42-yo male	Lentivirus	/	/	MPC	/	[84]
BrS2	ND	/	/	/	/	BrS	67-yo male	Lentivirus	/	/	MPC	/	
BrS3	ND (+*CACNA1C* Benign)	int19 position-7	/	/	/	BrS	24-yo female	Lentivirus	/	/	MPC	/	
LQTS3/BrS iPSCs	*SCN5A*	c.5349G>A	/	p.E1784K	Het	BrS/LQTS3	20-yo male	Retrovirus	HDAdV	corrected-LQTS3/BrS iPSCs	MPC; MEA	/	[64]
									Lentivirus (siRNA 4392420 for SCN3B)	LQTS3/BrS SCN3B siRNA	MPC		
BR1-P3M	*PK2P*	c.302G	/	p.R101H	Het	BrS	42.4 ± 12.9 years	Lentivirus	/	/	MPC; MEA	Ajmaline (100 μM)	[6]
BR2-P5M	ND	/	/	/	/	BrS	42.4 ± 12.9 years	Lentivirus	/	/			
BR3-P6M	ND	/	/	/	/	BrS	42.4 ± 12.9 years	Lentivirus	/	/			
BrS1	*SCN5A*	c.677 C>T; c.4885 C>T	/	p. A226V; p.R1629X	Homo; Het	BrS	/	Synthetic modified mRNA	/	/	MPC	4-Aminopyridine (4-AP) (5 mM); Flecainide (10 mM)	[89]
BrS2	*SCN5A*	c.R1232W	/	p.T1620M	Het	BrS	Male	/	Genome Edited	/	MPC		
Patient-specific iPS	*SCN5A*	c.1100G>A	/	p.R367H9	Het	BrS	69-yo male	Episomal	/	/	MPC	/	[90]
BrS1 iPSC	*RRAD*	G>A	/	p.R211H	Het	BrS	41-yo male	/	/	/	MPC	/	[76]
BrS2 iPSC	*RRAD*	G>A	/	p.R211H	Het	BrS	Female	/	/	/	MPC	/	
Rad R211H ins	*RRAD*	G>A	/	p.R211H	Het	BrS	/	/	CRISPR-Cas9	/	MPC	/	
UMGi128-A (isBrSc2)	*SCN1B*	c.629T>C; c.637C>A	/	p.L210P; p.P213T	Het; Het	BrS	48-yo male	Sendai Virus	/	/	MPC	/	[77]
MUSIi009-A-1 (p.A735V-SCN5A hiPSC_4.3)	*SCN5A*	c.2204 C>T	/	p.A735V	Homo	BrS	Foetal male	Retrovirus	CRISPR/Cas9	/	/	/	[91]
											MPC	/	[66]
GOEi098-A (isBrSd1)	*SCN10A*	c.3803G>A; c.3749G>A	/	p.R1268Q; p.R1250Q	Het; Het	BrS	52-yo male	Sendai Virus	/	/	MPC	Ajmaline (3 µM; 10 µM; 30 µM)	[72]
									/	/	MPC	LPS (2 µg/mL; 8 µg/mL); NAC (1 mM); Chelerythrine (5 µM); H_2_O_2_ (200 µM)	[73]
MMRL1126	*SCN9A*; *FKBP1B*; *PXDNL*	c.3253T>C; c.136delC; c.1172G>A	/	p.S1085P; p.P46L fsx22; p.R391Q	Het; Het; Het	BrS	Male	/	/	/	MPC	/	[78]
SCVIi026-A	*SCN5A*	c.53506 G>A	/	p.E1783K	Het	BrS	52-yo female	Sendai Virus	/	/	/	/	[78]
SCVIi027-A	*SCN5A*	c.2102 C>T	/	p.P701L	Het	BrS	36-yo male	Sendai Virus	/	/	/	/	
BrS1-iPSCs	*SCN5A*	c.C5435A	/	p.S1812X	Het	BrS	50-yo male	Lentivirus	/	/	APC; MEA	Cilostazol (10 µM); Milrinone (2.5 µM)	[92]
BrS2-iPSCs	*SCN5A*	c.C5435A	/	p.S1812X	Het	BrS	Female	Sendai Virus	/	/	APC; MEA		
BJTTHi001-A-2	*SCN1B*-KO	/	/	/	Homo	BrS/ERS	36-yo male	Episomal	CRISPR-Cas9 mediated gene KO	/	/	/	[93]
BBANTWi006-A (BrS9 C7)	*SCN5A*	c.4813 + 3_4813 + 6dupGGGT	/	/	Het	BrS	50-yo male	Sendai Virus	/	/	/	/	[93]
BBANTWi007-A (BrS10 C3)	*SCN5A*	c.4813 + 3_4813 + 6dupGGGT	/	/	Het	BrS	46-yo female	Sendai Virus	/	/	/	/	
IDIBGIi002-A (Rb20234)	*SCN5A*	c.4573G>A	/	/	Het	BrS	14-yo female	Episomal	/	/	/	/	[94]
IDIBGIi003-A (Rb20235)	*SCN5A*	c.4573G>A	/	/	Het	BrS	11-yo male	Episomal	/	/	/	/	
IDIBGIi004-A (Rb20236)	*SCN5A*	c.4573G>A	/	/	Het	BrS	45-yo female	Episomal	/	/	/	/	
SCN5A c.4437 + 5G>A iPSC	*SCN5A*	c.4437 + 5G>A	/	/	/	BrS/LQTS3	Healthy Patient	/	CRISPR/Cas9	/	/	/	[3]
isBrSb2	*CACNB2*	c.425C>T	/	p.S142F	Het	BrS	55-yo male	Sendai Virus	CRISPR/Cas9	isBrSb2-corr	MPC	Bisoprolol (30 nM; 300 nM; 3000 nM); Quinidine (10 µM)	[70]
KCND3-V392I-derived iPSC	*KCND3*	/	/	p.V392I	Homo	BrS/ERS (JWS)	18-yo male	Electroporation-based transfection method	CRISPR/Cas9	Isogenic-control iPSC	MPC; MEA	Acacetin (7.5 µM)	[74]
EK-iPSC	*CACNA1C*	c.3343 G>A	/	p.E1115K	Het	BrS/LQTS	12-yo male	Episomal	CRISPR/Cas9	GC-iPSC	MPC; MEA; Optical APs Recordings	Nifedipine (10 nM); Mexiletine (10 mM); GS-458967 (500 nM)	[71]
TBX5^G145R/WT^ iPSC—III.17	*TBX5*	c.G433A	/	p.G145R	Het	BrS	Female	Electroporation Episomal	/	/	MPC	/	[81]
TBX5^G145R/WT^ iPSC—IV.7	*TBX5*	c.G433A	/	p.G145R	Het	BrS	Female	Electroporation Episomal	CRISPR/Cas9	Isogenic control-iPSC	MPC; MEA	Ranolazine (3 µM); PDGF (10 µM); PIP3; IGF1	
Isogenic control-iPSC	*TBX5*	/	/	/	/	/	Female		CRISPR/Cas9	TBX5^WT/WT^; SCN5A^+/−^	MPC	/	
Isogenic control-iPSC	*TBX5*	/	/	/	Het	HOS	Female		CRISPR/Cas9	TBX5^−/WT^ 16bp Deletion	/	/	
Isogenic control-iPSC	*TBX5*	/	/	/	Het	HOS	Female		CRISPR/Cas9	TBX5^−/WT^ 22bp Deletion	/	/	
Isogenic control-iPSC	*TBX5*	/	/	/	Het	HOS	Female		CRISPR/Cas9	TBX5^−/WT^ 1bp Frameshift Insertion	/	/	

## 4. APC in the Study of BrS

### 4.1. APC and Heterologous Systems in the Study of BrS

In recent years, the DMS approach has been used to evaluate the pathogenicity of *SCN5A* variants found in Brugada patients [29,95]. In a first seminal work, Glazer and colleagues used DMS to analyse 248 variants in the voltage sensor segment of the Na_V_1.5 channel. They were able to study up to 83 *SCN5A* variants by plating five different stable *SCN5A* variant-expressing cell lines at the time on a single 384-well APC chip [28]. At least two independent transfections and 10 individual cells were studied for each variant. Out of these 83 selected variants, 10 were previously described as control variants with a range of normal and abnormal channel functions, 10 were described as suspected benign variants never observed in BrS or LQT patients, and 63 were described as suspected BrS variants found at least in one case of BrS in the literature and they were relative rare or absent from the gnomAD database of putative controls [28]. The 10 control variants were used to assess the reliability of the system by comparing the peak current density values recorded by MPC from the literature to those recorded by APC. The remaining 73 variants were classified prior to the functional analysis by applying the American College of Medical Genetics and Genomics (ACMG) criteria. Of these, 61 were classified as VUS, 2 as likely benign and 10 as likely pathogenic. In the postfunctional APC-based evaluation, variants with a peak current density smaller than 50% of the WT or with a larger than 10 mV rightward shift in the half-maximal voltage for the activation were considered to have significant LoF, while variants with a peak current density higher than 75% and a late current larger than 1% were classified as abnormal GoF. These cut-off intervals were based on a previously established correlation between functional parameters and the risk associated with BrS and LQTS [95]. The results of the postfunctional APC analysis led to the reclassification of 80% VUS. In particular, 36 out of 61 were classified as likely pathogenic and 14 were classified as likely benign or benign; furthermore, 35 novel variants’ potential pathogenicity was identified. This research resulted in the identification and characterisation of 44 novel LoF variants, nearly doubling the number of known missense *SCN5A* variants with a peak current density less than 10% compared to that of the WT. In a second study, the same research team applied the DMS and APC approaches to evaluate the dominant negative effects of previously characterised *SCN5A* missense variants related to BrS [29]. The researchers expressed 35 LoF variants and 15 partial LoF variants in HEK293 cells in a heterozygous state and analysed them using 384-well APC chips. The LoF variants had a peak current density of less than 10% compared to the WT, while the partial LoF variants had a peak current density of between 10–50% compared to the WT. The results showed that nearly all LoF variants (32 out of 35) and less than half of the partial LoF variants (6 out of 15) displayed dominant negative behaviour. The authors’ detailed analysis of a large number of variants showed that dominant negative effects are not necessarily linked to a specific location within the protein. They also found that dominant negative variants of the *SCN5A* gene are more likely to result in BrS symptoms compared to other missense variants, with a 2.7-fold increased risk of symptomatic BrS compared to haploinsufficient variants [29]. Despite the limitations, such as the use of a heterologous expression system and the fact that variant classification does not entirely predict BrS risks in individual patients, these findings clearly emphasise the significance of the APC and the potential of a high-throughput approach in ion channel functional characterisation. One avenue for the further improvement of this approach could be the utilisation of BrS patient-derived hiPS-CMs or hiPS-CMs transfected with BrS-related ion channel variants as experimental models, instead of immortalised cell lines.

### 4.2. APC and hiPSC-CMs in the Study of BrS

In the perspective of personalised medicine strategies for BrS patients and fulfilling the third step of the CiPA initiative, it is imperative to incorporate the complex cellular modelling provided by hiPSC-CMs into APC. The main challenge in this regard is the requirement of a large (and costly) number of hiPSC-CMs and the difficulty in manually selecting individual clones. Traditionally, APC systems necessitate a minimum of 25,000 to 150,000 cells per well for a successful catch rate [96]. However, recent advancements in technology and protocols have reduced the cell density and volume needed, allowing for a successful catch rate of 75% with as few as 250 to 1000 cells per 5 µL well [97]. Another limitation of APC being applied to hiPSC-CMs is the impossibility of visually selecting individual cells. Nowadays, the efficiency of CMs’ differentiation protocols, which can reach up to 80% in purity [98], is suitable for MPC but may result in an average wastage of 20% of chip wells in APC. Thus, in order to avoid capturing noncardiomyocytes, various methods based on metabolic or surface marker selection have been employed [99]. Additionally, the use of blebbistatin (BB) has been recommended to improve the viability of hiPSC-CMs in suspension and maintain the integrity of their cell membrane. BB acts as an uncoupler of excitation–contraction and is commonly used to obtain viable native cardiomyocytes from Langendorff-perfused hearts or isolated cardiac tissue. By preventing hypercontraction and Ca^2+^ paradox-related injury, BB has been shown to enhance the viability of hiPSC-CMs and improve the quality of APC recordings [100]. As previously mentioned, the immaturity of hiPSC-CMs is a major limitation. However, thanks to technical advancements [65,66,67,68,69], derived cardiomyocytes have been used to study the pathophysiology of certain disease-related protein variants in APC systems. Li and colleagues derived iPSC-CMs from two BrS patients carrying the heterozygous point mutation p.S1812X in the Na_V_1.5 channel [92]. Both patients experienced severe arrhythmias and a first-degree atrioventricular block. Patient- and healthy control-derived iPSC-CMs were characterised using MPC and MEA, while for the assessment of potential pharmacological therapies, chip-based APC was employed. The authors found a significant reduction in Na_V_1.5 and connexin-43 protein expression in patient-derived iPSC-CMs, leading to slower conduction and increased beat-to-beat variability compared to healthy controls in MEA-recorded field potentials. The mutation also affected various ion channels; in fact, in addition to a noticeable decrease in the I_Na_ peak density, accompanied by a rightward shift in the activation curve, there was an increase in the I_to_ peak density and in the I_CaL_ window current, accounting for the slower maximal depolarisation velocity during spontaneous AP, prominent notch, and other AP irregularities such as EADs and delayed afterdepolarisations (DADs). APC was also used to study the pharmacological effects of two clinically used phosphodiesterase blockers, cilostazol and milrinone. These compounds are known to suppress hypothermia-induced ventricular tachycardia and ventricular fibrillation. Both cilostazol and milrinone reduced the incidence of proarrhythmic events in patient-derived iPSC-CMs by reducing the enhanced I_to_, without affecting I_Na_, suggesting their therapeutic potential in treating BrS patients.

To the best of our knowledge, this is the only published work in which the pathophysiological mechanism of a BrS-related *SCN5A* variant has been studied in hiPSC-CMs using APC.

## 5. Conclusions

The advancement of techniques such as the use of cardiomyocytes derived from patients and automatic patch-clamp technology in BrS are both contributing to significant progress in the field. Cardiomyocytes derived from patients provide a more accurate representation of human heart cells, allowing the study of disease mechanisms and the development of treatments in a more relevant and meaningful way. Meanwhile, high-throughput APC technology enables the rapid characterisation of electrical activity in multiple cells simultaneously, increasing the speed and efficiency of data collection. Moreover, it can be used to assess the effectiveness of new drugs by measuring their impact on the electrical activity of cells obtained from patients with this clinical condition. While MPC will always have a place in the cardiovascular research area, using both approaches with an eye on the future, investigators can gain a more comprehensive view of the underlying mechanisms of heart disease, and develop treatments that are better suited to the specific needs of individual patients.

## Figures and Tables

**Figure 1 ijms-24-06687-f001:**
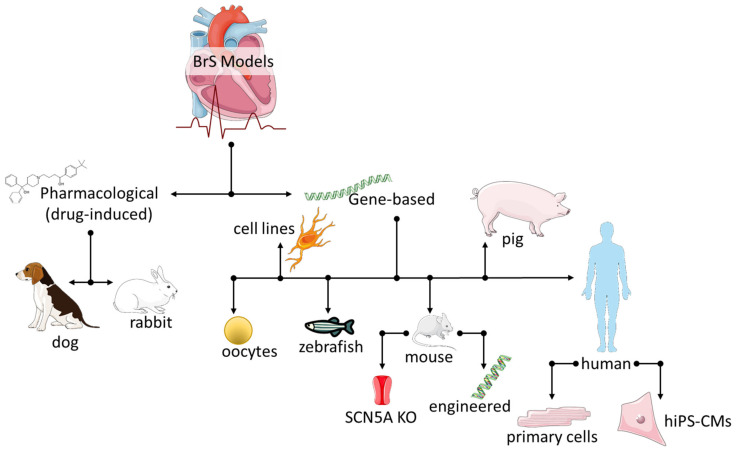
Experimental cellular and animal models developed throughout the years for the study of BrS-related protein variants with a BrS-like symptomology. The figure was partly generated using Servier Medical Art, provided by Servier, licensed under a Creative Commons Attribution 3.0 unported license.

**Table 1 ijms-24-06687-t001:** Automated patch-clamp applications in BrS-related ion channel variants.

Channel	Disorder	Ref.
KCNB1 (K_v_2.1)	Developmental and epileptic encephalopathy (DEE)	[22]
KCNB1 (K_v_2.1)	Neurodevelopmental disorder	[23]
KCNH2 (hERG)	Long-QT syndrome	[24]
KCNH2 (hERG)	Trafficking deficiency	[25]
KCNH2 (hERG)	Long-QT syndrome	[26]
KCNT1 (K_Na_1.1)	Infantile encephalopathy	[27]
KCNQ1 (K_v_7.1)	Congenital arrhythmias and Long-QT syndrome.	[18]
SCN5A (Na_v_1.5)	Brugada Syndrome	[28]
SCN5A (Na_v_1.5)	Brugada Syndrome	[29]

## Data Availability

No new data were created or analysed in this study. Data sharing is not applicable to this article.

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
