# Peer review of "Automated Patch-Clamp and Induced Pluripotent Stem Cell-Derived Cardiomyocytes: A Synergistic Approach in the Study of Brugada Syndrome"

_ijms, 2023, doi:10.3390/ijms24076687_

Round 1
Reviewer 1 Report
In this review, the authors summarized the experimental models and technologies used to study Brugada Syndrome (BrS). The authors reviewed the different BrS models including heterologous Expression cell models、animal models and human iPSC-derived cardiomyocytes models. They further talked about automated-patch clamp electrophysiology technology combined with hiPSC-CMs in the study of BrS. The combination of both technologies will significantly promote the research in the field. The readers can acquire most of the information in this review. I think the review is suitable for publication in the Journal.
minor comment:
Some abbreviations should be explained.
Author Response
The Authors thank the reviewer for the revision of the manuscript and the comment that gave us the opportunity to be more specific concerning the name of the genes and to explain abbreviations that were not included in the previous version. In the present version, the manuscript includes all the definitions of the abbreviations.
Please, see the attachment for the manuscript file with the correction required by Reviewer 1

Reviewer 2 Report
The review by Melgari , et al discusses the use of automated Patch-Clamp and induced Pluripotent Stem Cell-Derived Cardiomyocytes for the study of Brugada Syndrome. Overall this is a topic that may interest quite wide range of researchers in the field. It describes clearly the advances in the field and refer to many relevant articles.
I have one concern regarding the structure of the article. While the title focuses on the APC recordings and hiPSCs, I feel that much of the paper describes other, and not always relevant, approaches. For example, hiPSCs are first discussed only in page 6 and APC only on page 10 (from 14 text pages of the review). I think that some editing and shortening of the background part dealing with other approaches will improve the review, helping the reader focus on the main topic.
Another point is that is not clear to the reader why the authors chose to focus on BrS. The authors should clarify the importance of this disease and why such approaches are especially relevant for this disease.
Another general comment: the manuscript will benefit from some English editing. For example: Indeed, the alliance between hiPS-CMs and APC may represents a breakthrough in the study of a complex diseases as BrS (last sentence in the Introduction)
Author Response
We agree with the consideration about the usefulness to rethink at the structure of the manuscript. We anticipated the paragraphs related to the automated patch-clamp technique and shortening the paragraphs about the experimental models used in the BrS research. Indeed, this remodeling implied the reorganization of the bibliography.
We have improved the “Introduction” paragraph underlying the relevance of the BrS and why the automated patch clamp and the human induced pluripotent stem cell applications are strategic in the field.
Finally, we controlled English grammar.
Reviewer 3 Report
Dear Author,
Congratulation for your intellectual work in the field.
Authors have outlined the Automated patch-clamp and induced pluripotent stem cell-derived cardiomyocytes have shown great potential in the study of Brugada syndrome, a rare genetic disorder that affects the heart's rhythm and can lead to sudden cardiac death. These techniques offer several advantages over other modalities, including greater accuracy, higher throughput, and lower variability. Additionally, they are a useful tool for drug discovery and can help researchers better understand the underlying mechanisms of the disease. However, the economic cost of these technologies remains a concern, and more work is needed to evaluate their cost-effectiveness compared to other approaches.
Manuscript was well written. although, minor comments needs to be addressed .
Authors have described most of the up to date information on models. Although, its description in terms of central illustration (Figure) would be more interesting for readers perspective.
Conclusion is consistent with the evidence presented .
Author Response
The Authors thank the reviewer for the careful reading of the manuscript and for the kind words. We enthusiastically took the reviewer suggestion and included an illustration about the models described.We agree that this would attract more the attention of the reader, leading to a more clear and effective take home message.
Please find in the attachment the required figure
